## RESEARCH ARTICLE

# Nascent dendrite branches initiated by a localized burst of Spire-dependent actin polymerization

Deirdre Hatton[1,2,3,*], Claire Marquilly[1,2,*], Caitlin Hanrahan[1,2,3], Tiago Ferreira[1,2], Yimiao Ou[1,2], Lauren Cinq-Mars[1,2,3], Will Silkworth[4], Hannah M. Bailey[4], Margot E. Quinlan[4,5] and Donald J. van Meyel[1,2,‡]

## ABSTRACT

Dendrites form arbors whose size, shape and complexity define how neurons cover their receptive territories. Actin dynamics contribute to growth and remodeling of dendrite arbors. Here, we have examined how Spire, a conserved actin nucleation factor, promotes the formation of new branches *in vivo*. In live imaging of *Drosophila* class IV dendritic arborization (c4da) neurons, Spire was observed at new sites of branch initiation, where it assembled new actin polymer in a burst immediately before filopodial outgrowth. For dendrite arborization, Spire required intact structural domains to nucleate actin and target the secretory network, and interacted with Rab11 GTPase, a key regulator of recycling endosomes. Together, these findings support a model in which Spire cooperates with Rab11 to promote new dendrite branches by linking localized actin dynamics with intracellular trafficking of endosomes that deliver lipids and cargoes to fuel protrusive outgrowth of nascent dendrites.

KEY WORDS: Dendrite, Actin, *Drosophila*

## INTRODUCTION

Dendrites have distinct tree-like branching patterns that influence the number, strength and integration of sensory or synaptic inputs into neurons (Grueber et al., 2005; Lefebvre et al., 2015; London and Häusser, 2005). Understanding how dendrites are shaped during development is essential to gain full insight into the construction of neural circuits that underlie animal survival, behaviors, learning and memory (Luo, 2021). A major gap remains in our understanding of the spatial and temporal control of cytoskeletal remodeling responsible for dendrite morphogenesis during development (Lefebvre, 2021), though it is known to involve regulation of actin dynamics (Konietzny et al., 2017) by specific proteins to nucleate, elongate and remodel actin filaments (Rottner et al., 2017; Siton-Mendelson and Bernheim-Groswasser, 2017).

[1]Centre for Research in Neuroscience, Department of Neurology and Neurosurgery, McGill University, Montreal, Quebec, H3G 1A4 Canada. [2]Research Institute of the McGill University Health Centre, Montreal, Quebec, H3G 1A4 Canada. [3]Integrated Program in Neuroscience, McGill University, Montreal, Quebec, H3A 2B4 Canada. [4]Department of Chemistry and Biochemistry, University of California Los Angeles, Los Angeles, CA 90095, USA. [5]Molecular Biology Institute, University of California Los Angeles, 607 Charles E. Young Drive, Los Angeles, CA 90095, USA.
*These authors contributed equally to this work

‡Author for correspondence (don.vanmeyel@mcgill.ca)

 D.J.v.M., 0000-0002-6075-8599

Furthermore, to cope with their size and complexity, dendrites have specialized arrangements of secretory network organelles for the biogenesis of cargo-carrying vesicles and their transport to the cell surface (Kennedy and Hanus, 2019). Given the importance of actin and the secretory network for dendrite development and function, it is important to clarify how they are coordinated to initiate new dendrite branches *in vivo* (Lanoue and Cooper, 2019; Lefebvre, 2021).

New dendrite branches arise from actin-rich filopodia that rapidly extend from initiation sites along established branches (Lanoue and Cooper, 2019). Dendrite filopodia are structurally distinct from conventional filopodia, with an unusual network-like cytoskeletal organization characterized by both branched and linear filaments of mixed polarity (Korobova and Svitkina, 2010), suggesting involvement of multiple actin regulatory proteins with distinct activities. Few studies have identified specific actin regulators involved in forming new branches (Hou et al., 2015; Nithianandam and Chien, 2018; Shi et al., 2021; Stürner et al., 2022, 2019; Zou et al., 2018), and none have pinpointed their regulation of actin to the location and timing of nascent branch outgrowth. In *Drosophila* larvae, dendritic arborization (da) neurons provide an excellent model with which to study the roles of actin regulators in the development of dendrite arbors *in vivo* (Grueber et al., 2002; Medina et al., 2006; Nagel et al., 2000). In da neurons, the locations of dendrite filopodia initiation sites are pre-figured by patches of actin polymer (Andersen et al., 2005; Nithianandam and Chien, 2018; Stürner et al., 2019), as has also been proposed for dendrite filopodia in the mouse brain (Willig et al., 2014). For the large space-filling arbors of class IV da (c4da) neurons, the formation of new dendrites requires the Arp2/3 actin nucleator complex under the control of the activator WAVE and the small GTPase Rac1 (Stürner et al., 2019). In c4da neurons, new dendrite formation also involves F-actin severing by Twinstar/Cofilin in the production of 'blobs' of pre-assembled actin polymer, which move bi-directionally within dendrites, pause before filopodium outgrowth and appear to contribute F-actin into the nascent branch (Nithianandam and Chien, 2018). The factors specifying the location and timing of filopodial outgrowth from actin patches and blobs have not been defined.

Here, we explored the role of Spire (Spir), a conserved multi-domain actin nucleation factor. We have previously shown that knockdown of Spir with RNA interference (RNAi) in cd4da neurons reduces the number of terminal dendrite branches and compromises the ability of these nociceptive neurons to elicit larval escape from noxious stimulus (Ferreira et al., 2014a). Spir is known to cooperate with Fmn-family formins, which are also actin nucleators, during *Drosophila* and mouse oogenesis. Spir has four WASP homology 2 (WH2)-domains for actin nucleation, a KIND domain for interaction with Fmn-family formins (Manseau and Schüpbach, 1989; Pfender et al., 2011; Quinlan et al., 2005; Vizcarra et al., 2011), a modified FYVE (mFYVE) domain to target vesicles of the

secretory network and a SPIR-box with potential to interact with Rab GTPases (Kerkhoff et al., 2001; Quinlan et al., 2005, 2007; Tittel et al., 2015; Yao et al., 2024), which play roles in vesicle sorting and transport (Kiral et al., 2018; Welz and Kerkhoff, 2023). Previous studies showed functional links between Spir and Rab11 in oocytes, in cultured cells and in vitro (Kerkhoff et al., 2001; Pylypenko et al., 2016; Schuh, 2011; Yao et al., 2024), and Rab11 has been shown to be required for correct dendrite arborization of c4da neurons (Krämer et al., 2019; Lin et al., 2020; Mitchell et al., 2023). Here, we show that Spir stimulates a burst of localized actin polymerization immediately before filopodial outgrowth of new dendrite branches and that Spir cooperates with Rab11 in dendrite arborization. Our data support a model in which Spir and Rab11 link localized actin assembly with the secretory network to deliver lipids and cargoes to nascent dendrites.

## RESULTS
### Spir is required in c4da neurons for localized F-actin synthesis immediately before outgrowth of new dendrite branches

Development of *Drosophila* c4da dendrites begins in embryogenesis and continues in larval stages, with arbors growing in the larval body wall between muscles and epithelial cells (Grueber et al., 2002). We produced a *GAL4* line inserted in the *spir* locus (*spir^GAL4^*) and used it to drive the reporter *UAS-mCD8::GFP. spir^GAL4^* was expressed in c4da neurons (marked by ppk-CD4::tdTom) and other sensory neurons in the larval body wall (Fig. 1A), consistent with our previous report (Ferreira et al., 2014a). To examine Spir protein expression within c4da neurons, we developed new anti-Spir sera for immunohistochemistry (IHC) (Fig. S1A), and we also introduced an in-frame epitope-tag (smGFP-HA) into the *spir* locus (Fig. S1C,D). Both approaches showed Spir to be expressed in c4da neurons in third instar (L3) larvae.

To reveal actin dynamics leading up to the protrusion of dendrite filopodia, we acquired time-lapse movies in second instar (L2) larvae, capturing F-actin (labeled with LifeAct::GFP) within c4da neurons (labeled with ppk-CD4::tdTom). To study how loss of *spir* affects actin accumulation and new branch initiation in c4da neurons, we used the mutant alleles *spir^1^* and *spir^2F^*, which prematurely truncate Spir within the KIND domain (Fig. 1B) (Wellington et al., 1999). We examined each allele in combination with a deficiency for *spir* (*spir^Df^*), which has a distinct genetic background. All heteroallelic combinations of these alleles of *spir* are viable, and so with anti-Spir sera we were able to confirm that *spir* mutants lack Spir expression in larval sensory neurons (Fig. S1B) and adult brain lysates (Fig. S1E).

We captured movies of developing c4da dendrites in L2 larvae and mapped the appearances of new branches in control animals heterozygous for *spir^Df^* (Fig. 1C,D, Movies 1 and 2) or *spir^2F^/+* (Fig. 1F), or in wild type (+/+, Fig. 1F). In all of these controls, LifeAct::GFP was seen in stationary puncta that appeared prior to branch initiation; in wild type, (+/+) approximately half of these puncta appeared *de novo* (Fig. 1C,F, Movie 1) and the other half were pre-formed actin blobs that moved to the site (Fig. 1D,F, Movie 2), consistent with a previous report (Nithianandam and Chien, 2018). In *spir^2F^/spir^Df^* mutants, however, new branches arose only from pre-formed actin blobs that moved to the initiation site and not from stationary actin puncta produced *de novo* (Fig. 1E,F, Movie 3).

In controls where branches initiated from either blobs or stationary puncta, LifeAct::GFP signal at these sites became more intense immediately before branch initiation and then extended into the protruding nascent branchlet (Fig. 1C,D). We quantified this change of intensity in wild-type (+/+) controls, where LifeAct::GFP

increased roughly twofold on average in the 60 s prior to outgrowth and remained at this intensity for ≥60 s (Fig. 1G). In contrast, in *spir^2F^/spir^Df^* mutants the LifeAct::GFP signal in blobs did not increase prior to branch initiation (Fig. 1E,G), indicating that Spir is required for this localized assembly of F-actin immediately before the outgrowth of new dendrite branches.

### Localization of Spir at nascent branches
In c4da dendrites, Spir was observed in puncta in dendrite branches near the cell body, but strong labelling from epithelial cells and muscles prevented their detection and quantification in more distal branches (Fig. S1A,C,D). To overcome this, we expressed a *UAS-Spir::GFP* transgene (Fig. 1B) in c4da neurons with *ppk-GAL4* and characterized the distribution of Spir::GFP (Fig. S1F). In L3 larvae, GFP-labeled puncta were seen in 100% of shafts, 28% of branch points and 6% of branch tips (Fig. S1G). Consistent with this, most of these GFP-labeled puncta were found along shafts (57%) or at branch points (36%) (Fig. S1G).

We next used live imaging to determine whether Spir::GFP was localized to sites of new branch initiation in L2 larvae. Movies (~1 frame/2 s for >5 min) showed a distribution of Spir::GFP (Fig. 2A,B) similar to that of fixed larvae at L3 (Fig. S1F). Some puncta were stationary while others were steadily motile or moved in a saltatory manner with occasional pauses (Fig. S1H, Movie 4). Motile puncta were often smaller and rounder, while stationary puncta were often larger and irregular in shape. Importantly, every nascent branch in controls (+/+, L2) had a punctum stationed precisely at (68%) or near (<2 μm, 32%) the initiation site prior to outgrowth (Fig. 2C). These puncta arrived from elsewhere (30%, Fig. 2A,D, Movie 5) or were already at that site in the first frame (70%, Fig. 2B,D, Movie 6), and the time until branch outgrowth varied (Fig. 2D). Spir::GFP intensity did not typically increase prior to outgrowth but, in some instances (12%), a motile punctum combined with one already stationed at the future initiation site. Spir::GFP at branch initiation sites was stably maintained throughout the process of filopodial outgrowth, stabilization or retraction, and invaded the filopodia as they grew (Fig. 2B, Movie 6).

### Spir is required for new branch formation
At the outset of our timelapse movies, the number of terminal dendrites/μm² was reduced in *spir^2F^/spir^Df^* mutants compared to controls and, importantly, so was the number of new branches/μm² formed per min during each movie (Fig. 2E). This was also observed in movies of c4da neurons expressing LifeAct::GFP instead of Spir::GFP (Fig. S2A,B). For new branches, growth occurred by saltatory extensions in both controls and *spir^2F^/spir^Df^* mutants, but there were no differences between them in mean speed of outgrowth, proportion of time spent paused between extensions or maximum length of protrusion (Fig. S2C-E). For pre-existing branches, the proportion that were static increased in *spir^2F^/spir^Df^* mutants (Fig. S2F-I), whether LifeAct::GFP was present or not. Together, these results confirm that Spir promotes the formation of new branches and contributes to the dynamics of pre-existing branches.

Consistent with this, we examined the consequences of Spir function for fully formed arbors of c4da neurons at L3 and found that *spir* mutants (*spir^1^/spir^Df^* or *spir^2F^/spir^Df^*) had fewer terminal branches than controls (*spir^2F^/+*) (Fig. 3A,C), and thus shorter total arbor length (Fig. 3D). Sholl analysis also showed reduced peaks of maximum branch density (critical value) (Fig. 3B,F). Dendritic field was reduced in some c4da neurons of *spir^2F^/spir^Df^* mutants (Fig. 3E), and so we repeated the analysis after removing a subset (4/15) of cells with unusually small dendritic fields (Fig. S3A-D). As expected, we found that dendritic field was no longer

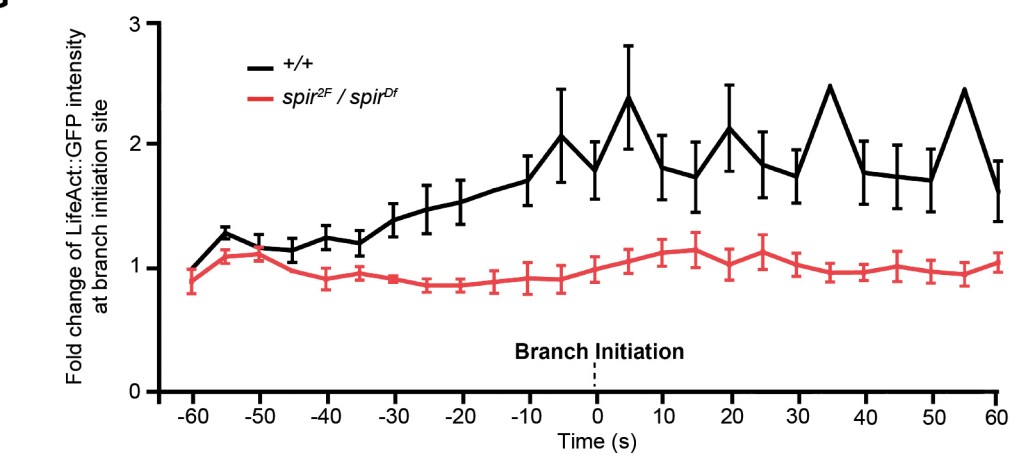

**F**

| Nascent Dendrites with: | Occurrence (Mean (%) ±sd) | | |
|---|---|---|---|
| | **+/+** | *spir²ᶠ/⁺* | *spir²ᶠ/ᴰᶠ* |
| *De novo* F-actin | 50±19 | 29±37 | 0 ± 0 |
| Pre-formed F-actin | 49±20 | 71±36 | 100 ± 0 |

Fig. 1. See next page for legend.

**Fig. 1. Spir is required for localized F-actin synthesis immediately before new branch outgrowth.** (A) spir$^{GAL4}$-driven mCD8::GFP in c4da neurons (ppk-CD4::tdTom), including ddaC (arrowheads), and other sensory neurons in an L3 larva. Scale bar: 25 μm. (B) Diagram showing Spir protein domains (KIND, WH2, SB and mFYVE), the locations of spir$^{2F}$ and spir$^1$ mutations, and the names of the disruptive mutations (Y232K, ABCD, ΔSB and C5,6S) introduced into Spir::GFP. (C,D) Filmstrips (Movies 1 and 2) of new dendrite branch formation in a c4da neuron (ppk-CD4::tdTom) from a control (spir$^{Df}$/+) L2 larva. LifeAct::GFP (driven by ppk-GAL4) reveals actin polymer within dendrites. Arrows in C indicate a punctum of F-actin appearing *de novo*; arrowheads in D indicate a pre-existing punctum that moved from its original position (white arrowheads) to the site of branch initiation (green arrowheads). In all instances, LifeAct::GFP signal intensified immediately before branch initiation. Scale bars: 2 μm. (E) Filmstrip (Movie 3) showing new dendrite branch formation in a spir$^{2F}$/spir$^{Df}$ larva at L2, where LifeAct::GFP moved to the initiation site but did not intensify prior to outgrowth (arrow). Scale bar: 2 μm. (F) The proportion of newly formed terminal branches in L2 larvae that had either *de novo* F-actin or pre-formed F-actin prior to branch initiation in c4da neurons. Data are mean±s.d., comparing controls (+/+, n=50 new branches in movies from 11 neurons) with spir heterozygotes (spir$^{2F}$/+, n=18 new branches, 8 movies) and spir mutants (spir$^{2F}$/spir$^{Df}$, n=8 new branches, 11 movies). (G) Fold-change of F-actin intensity (LifeAct:GFP) at branch initiation sites, from 60 s before branch formation to 60 s afterwards, comparing mean±s.e.m. of controls (black) and spir mutants (red).

significantly different from controls (Fig. S3C) but the branching deficits were still strongly evident for the number of terminal dendrites, arbor length and critical value (Fig. S3A,B,D).

Importantly, the selective expression of UAS-spir::GFP in c4da neurons with ppk-GAL4 (Fig. 3A) fully rescued the number of terminal branches in spir$^{2F}$/spir$^{Df}$ mutants (Fig. 3C, Fig. S3A) and

critical value (Fig. 3B,F, Fig. S3D), demonstrating that Spir acts within c4da neurons to control these features. It did not fully rescue total arbor length (Fig. 3D, Fig. S3B), which is not likely to be due to inadequate expression since IHC (Fig. S1F) and western blots (Fig. S1E) showed Spir::GFP at higher levels than endogenous Spir. Perhaps additional cell types contribute to total arbor length, such as epithelial cells in the larval body wall where Spir is expressed (Fig. S1A-D). We conclude that Spir acts within c4da neurons to establish the correct number of terminal branches, likely by contributing to filopodial outgrowth necessary for nascent branches.

## Each domain of Spir is required for dendrite arborization

We sought to determine the importance of the four identified protein domains of Spir by testing if disruptive mutations in Spir::GFP (Fig. 1B) could rescue the correct numbers of terminal branches in c4da neurons of spir mutants. The Y232K mutation is a missense substitution in the KIND domain known to abolish the interaction between Spir and the Fmn-family formin Cappuccino (Capu) in oocytes (Quinlan, 2013; Vizcarra et al., 2011). The ABCD mutation is a series of four triple-alanine substitutions in the WH2 domains that greatly diminish the ability of Spir to polymerize F-actin *in vitro* (Quinlan et al., 2005). The ΔSB mutation is a 20 amino acid deletion of the entire Spir-box domain that has sequence similarity to the Rab effector protein Rabphilin3a, suggesting it interacts with Rab GTPases (Kerkhoff et al., 2001). Finally, the C5,6S mutation involves substitution of serine residues for two key cysteines in the so-called modified FYVE (mFYVE) domain. FYVE domains interact in lipid membranes with phosphatidylinositol 3-phosphate (PtdIns3P), a prominent component of the membranes in the

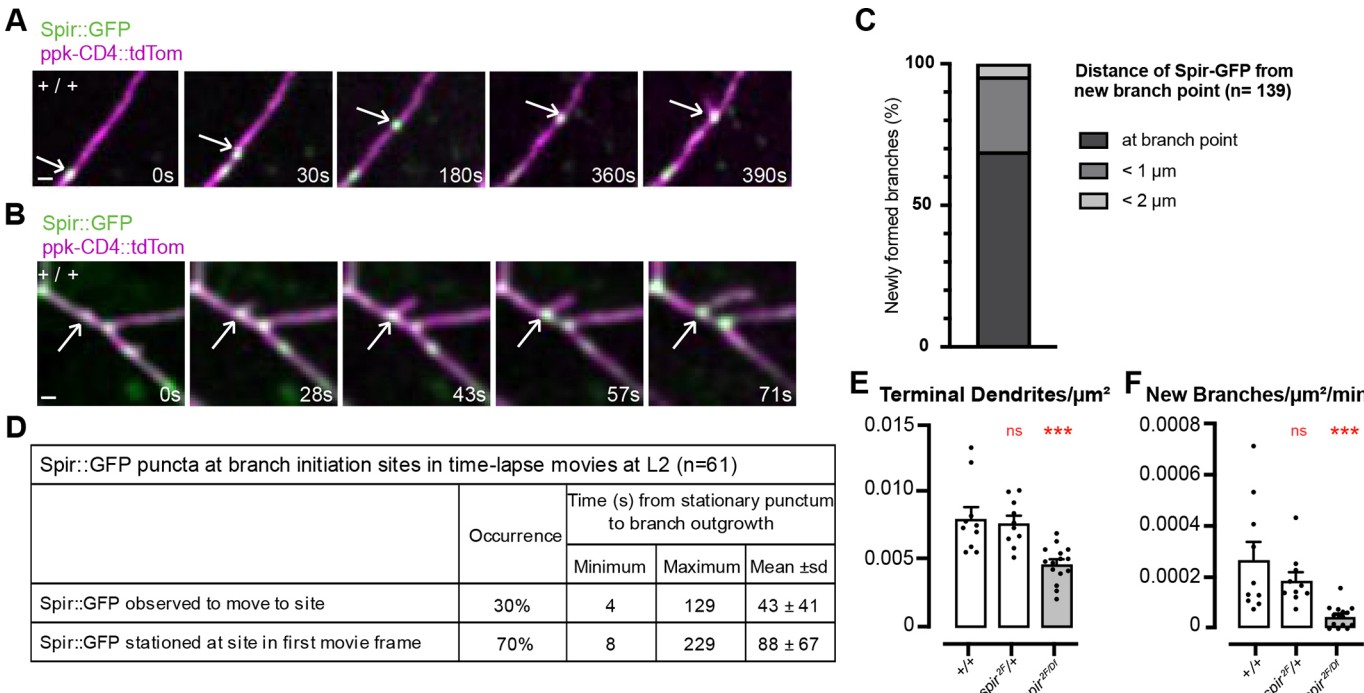

**Fig. 2. Spir is at initiation sites for new dendrite branches and is required for their formation.** (A,B) Filmstrips (Movies 5 and 6) of control (+/+) L2 larva, where Spir::GFP (arrows) was observed to move to the initiation site of a new dendrite branch (A) or was already stationed there at the beginning of the movie (B). Scale bars: 2 μm. (C,D) For newly formed dendrite branches, the distance of the closest Spir::GFP puncta from the site of branch initiation was measured (C), and the time from observation of a stationary Spir::GFP punctum at the site to filopodial outgrowth was recorded (D). (E,F) Dendrite parameters from time lapse movies of control (+/+, n=104 new branches, 10 movies), heterozygotes (spir$^{2F}$/+, n=91 new branches, 10 movies) or spir$^{2F}$/spir$^{Df}$ mutants (n=42 new branches, 15 movies). Asterisks indicate significant changes compared to +/+ wild-type controls (left-most white bars), as described in Materials and Methods (Graphs and Statistics) (***P<0.001; ns, not significant). (E) Terminal branches/μm$^2$ in first movie frame [ANOVA $F_{(2, 32)}$=12.30, P<0.0001]. (F) Number of new branches/μm$^2$/min in each movie [ANOVA $F_{(2, 32)}$=9.276, P=0.0007].

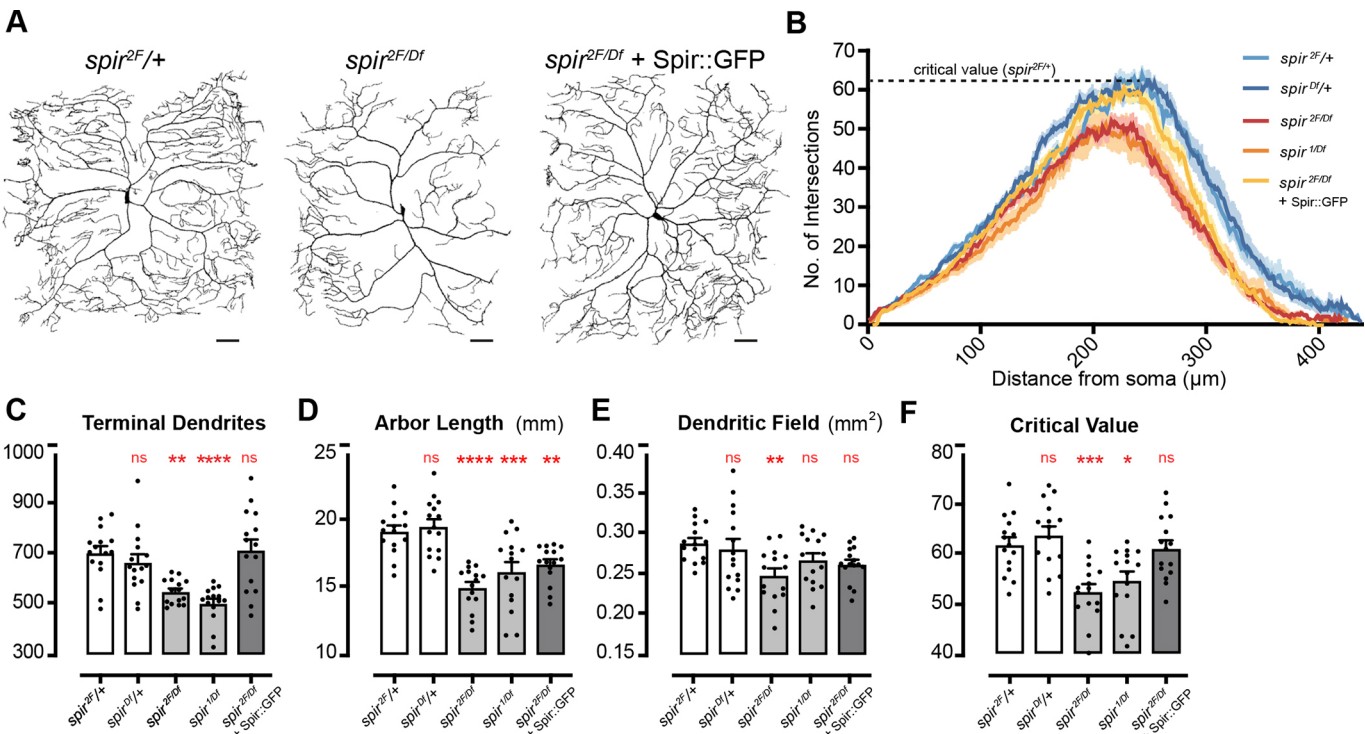

**Fig. 3. Spir is required for dendrite arborization of c4da neurons.** (A) Dendrite arbors in a *spir* heterozygote control (left), a *spir* mutant (middle) and a *spir* mutant rescued with Spir::GFP (right). Scale bars: 50 µm. (B) Sholl profiles with dotted line indicating critical value for controls ($spir^{2F/+}$). (C-F) Dendrite parameters per neuron in L3 larvae (*n*=15 for each genotype). Asterisks indicate significant changes compared to $spir^{2F/+}$ heterozygous controls (left-most white bars) (*$P<0.05$, **$P<0.01$, ***$P<0.001$, ****$P<0.0001$; ns, not significant). (C) Total number of terminal dendrites per neuron [ANOVA $F_{(4, 70)}=11.30$, $P<0.0001$]. (D) Total length of dendrite arbors (mm) [ANOVA $F_{(4, 70)}=15.12$, $P<0.0001$]. (E) Dendritic field area (mm$^2$) [ANOVA $F_{(4, 70)}=3.352$, $P=0.0144$]. (F) Sholl critical value [ANOVA $F_{(4, 70)}=8.936$, $P<0.0001$].

endosomal system (Marat and Haucke, 2016). The mFYVE domain in Spir binds less specifically to negatively charged lipids, including PtdIns3P, and is required for the localization of Spir to intracellular membranes (Tittel et al., 2015).

When expressed in c4da neurons (with *ppk-GAL4*) of $spir^{2F}$/$spir^{Df}$ mutants, none of these mutations rescued the number of terminal dendrites per neuron (Fig. 4A) or critical value (Fig. 4B, Fig. S4A). This was not due to insufficient supply, because none were expressed at lower levels than intact Spir::GFP in western blots of adult head lysates (Fig. 4C,D and Fig. S4B). The Y232K, ABCD and C5,6S mutations did not obviously affect the punctate distribution of Spir::GFP in static images of c4da arbors in L3 larvae (Fig. S4C), but the ΔSB mutation caused it to accumulate near the cell soma and form a more continuous pattern (Fig. 4E). Though this might be explained in part by unusually high expression levels (Fig. 4C,D and Fig. S4B), these results suggest that the SB domain is required for proper localization of Spir within c4da arbors. None of the mutants appear to have dominant-negative activity, since expression in a wild-type (+/+) background caused no significant differences in the number of terminal dendrites or total arbor length compared to Spir::GFP (Fig. S5A-E), though the ΔSB mutation trended toward reductions in both parameters. Based on the failure of these four mutations to rescue $spir^{2F}$/$spir^{Df}$ mutants, we conclude that each domain of Spir contributes to its control of dendrite branching.

## Spir and Rab11 cooperate for dendrite arborization

The pattern and motility of Spir::GFP puncta within dendrites of c4da neurons was reminiscent of elements of the secretory network, where Rab GTPases, including Rab11, are key regulators of vesicle

and cargo trafficking (Kiral et al., 2018). Rab11 is associated with the trans-Golgi network, post-Golgi and recycling endosomes (Sonnichsen et al., 2000). Rab11 has been shown to be required for correct c4da dendrite arborization (Krämer et al., 2019; Lin et al., 2020; Mitchell et al., 2023), a finding we confirmed with RNAi knockdown of *Rab11* mRNA in c4da neurons (Fig. 5A), which reduced the numbers of terminal dendrites in L3 larvae (Fig. 5B), total arbor length (Fig. 5C), dendritic field (Fig. 5D) and Sholl critical value (Fig. 5E,F) compared to controls (+/+). In contrast, overexpression of Rab11::GFP in c4da neurons was not sufficient to affect the number of terminal dendrites, arbor length and Sholl critical value at L3 (Fig. S6A,B,D), though the dendritic field was smaller (Fig. S6C).

Whether Rab11 is involved in the initiation of new dendrite branches has not been determined. To address this, we mapped the appearance of new branches in movies (~1 frame/2 s for >5 min) of developing c4da dendrites in L2 larvae (Fig. 5G, Movie 7). We began by examining the distribution of Rab11::GFP, the expression of which had no effect on the dynamics of pre-existing branches (Fig. S6E-H). In controls (+/+, L2) we observed Rab11::GFP at (60%) or near (<3 µm, 40%) the initiation site of each new dendrite branch (Fig. 5H). Some Rab11::GFP puncta were motile within arbors, but for every nascent branch we observed (*n*=48), Rab11::GFP was already stationed at the initiation site from the beginning of every movie and so we could not determine how long it had been there prior to outgrowth. Depletion of Rab11 by RNAi knockdown reduced the total number of terminal branches/µm$^2$ at L2 (Fig. 5I) and, importantly, severely reduced the number of new branches/µm$^2$/min in each movie (Fig. 5J), indicating that Rab11 is indeed required for the initiation of new dendrite branches. As in *spir*

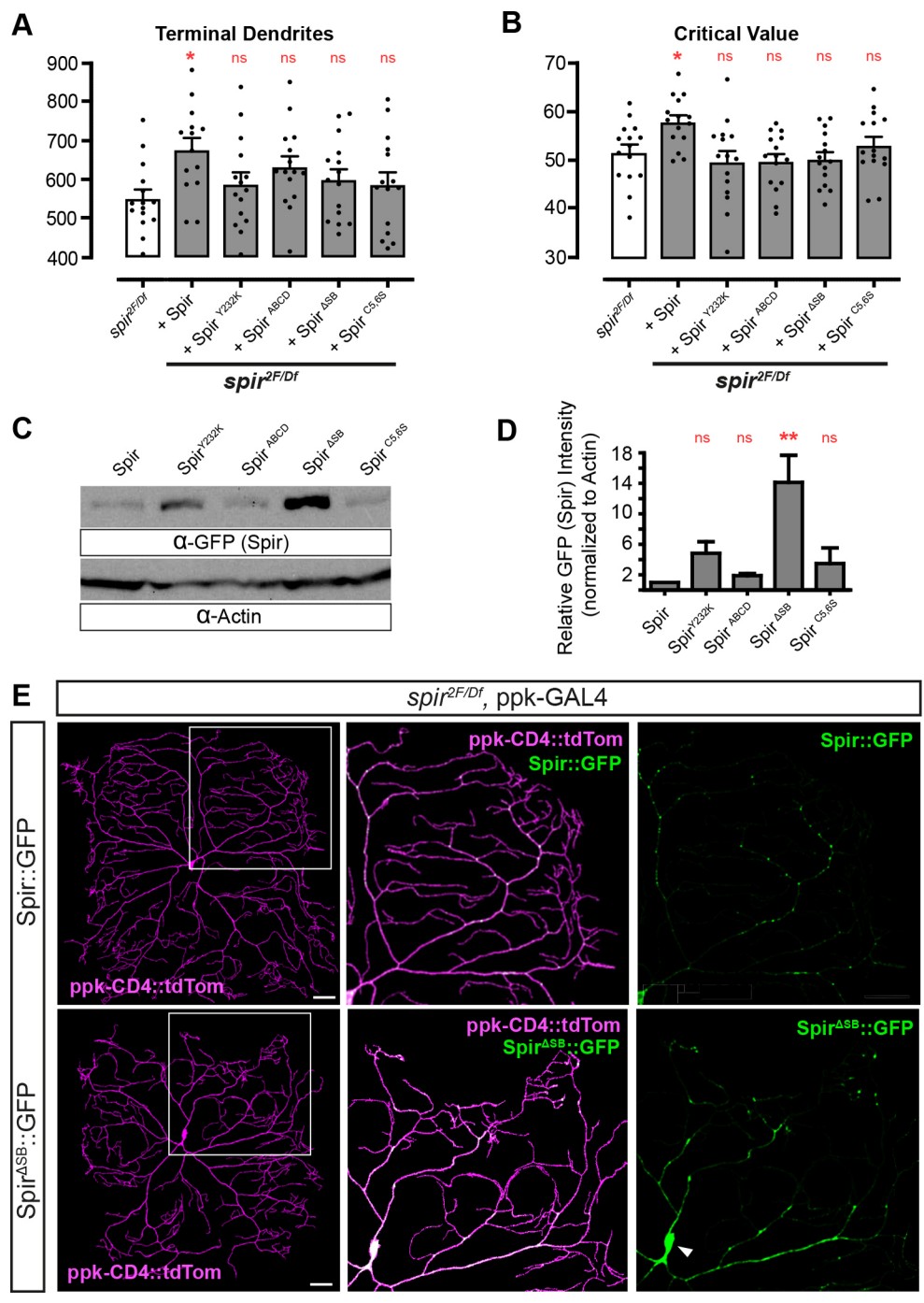

**Fig. 4. Structure-function analysis to identify Spir domains required for dendrite arborization and Spir localization.** (A,B) Dendrite parameters (L3 larvae) showing rescue of $spir^{2F}/spir^{Df}$ mutants with Spir::GFP, but failure to rescue by domain-specific mutations ($n$=15 for each genotype). Asterisks indicate significant changes compared to $spir^{2F/Df}$ mutant controls (white bars) (*$P$<0.05, **$P$<0.01; ns, not significant). (A) Total number of terminal dendrites per neuron [ANOVA $F(5, 83)$=2.269, $P$=0.0550]. (B) Sholl critical value [ANOVA $F(5, 84)$=3.552, $P$=0.058]. (C) Anti-GFP and anti-Actin western blot analysis from adult head lysates, where transgenes were expressed in all neurons with nSyb-GAL4. Expected size of Spir::GFP is 142 kDa, and endogenous Actin is 42 kDa. (D) Relative band intensities on western blots ($n$=3) of Spir::GFP constructs (anti-GFP) normalized to anti-Actin. Asterisks indicate that Spir$^{\Delta SB}$::GFP is expressed at significantly higher levels than Spir::GFP [Friedman test Q (4)=10.40, $P$=0.0053, with Dunn's post-hoc tests]. (E) ppk-GAL4-driven Spir::GFP or Spir$^{\Delta SB}$::GFP in a spir mutant background. Areas outlined in the left panels are magnified in the panels on their right. Scale bars: 50 μm. Spir::GFP is punctate in dendrites, but Spir$^{\Delta SB}$::GFP accumulates in a continuous pattern in dendrites proximal to the cell soma (white arrowhead).

mutants (Fig. S2F-I), pre-existing branches were also affected by Rab11 knockdown (Fig. S6E-H): the proportion of branches that both extended and retracted was reduced (Fig. S6G), while the proportion of static branches increased (Fig. S6H). Therefore, loss of Rab11 results in fewer new branches and reduced dynamics of existing ones.

Since Rab11, like Spir, is required for proper dendrite arborization, is localized to sites of new branches and is required for their initiation, we wondered how Rab11 and Spir might be related in regulating these processes. We first asked whether expressing excess Spir::GFP could restore dendrite arborization and new branch formation to neurons lacking Rab11. We found that it did not (Fig. 5B-F,I,J). In the converse experiment, Rab11::YFP overexpression did not restore dendrite arborization to spir mutants (Fig. S6I-M). Together these findings suggest that Spir and Rab11

do not lie in a simple linear pathway controlling dendrite development, and so we wondered whether Spir and Rab11 might interact and thereby cooperate with one another for dendrite arborization instead.

Genetic interaction experiments that examine trans-heterozygotes for mutant alleles of spir and Rab11 can provide evidence that they cooperate in a common molecular pathway. When compared to control heterozygotes for either mutation alone (Fig. 6A) the c4da neurons of trans-heterozygotes showed reduced numbers of terminal branches (Fig. 6B) and arbor length (Fig. 6C), while dendritic field was largely unaffected (Fig. 6D).

We next tested whether Spir and Rab11 can physically associate with one another in vivo, by co-expressing Spir-tdTom (in all neurons, with nSyb-GAL4) either with Rab11::GFP or Capu::GFP

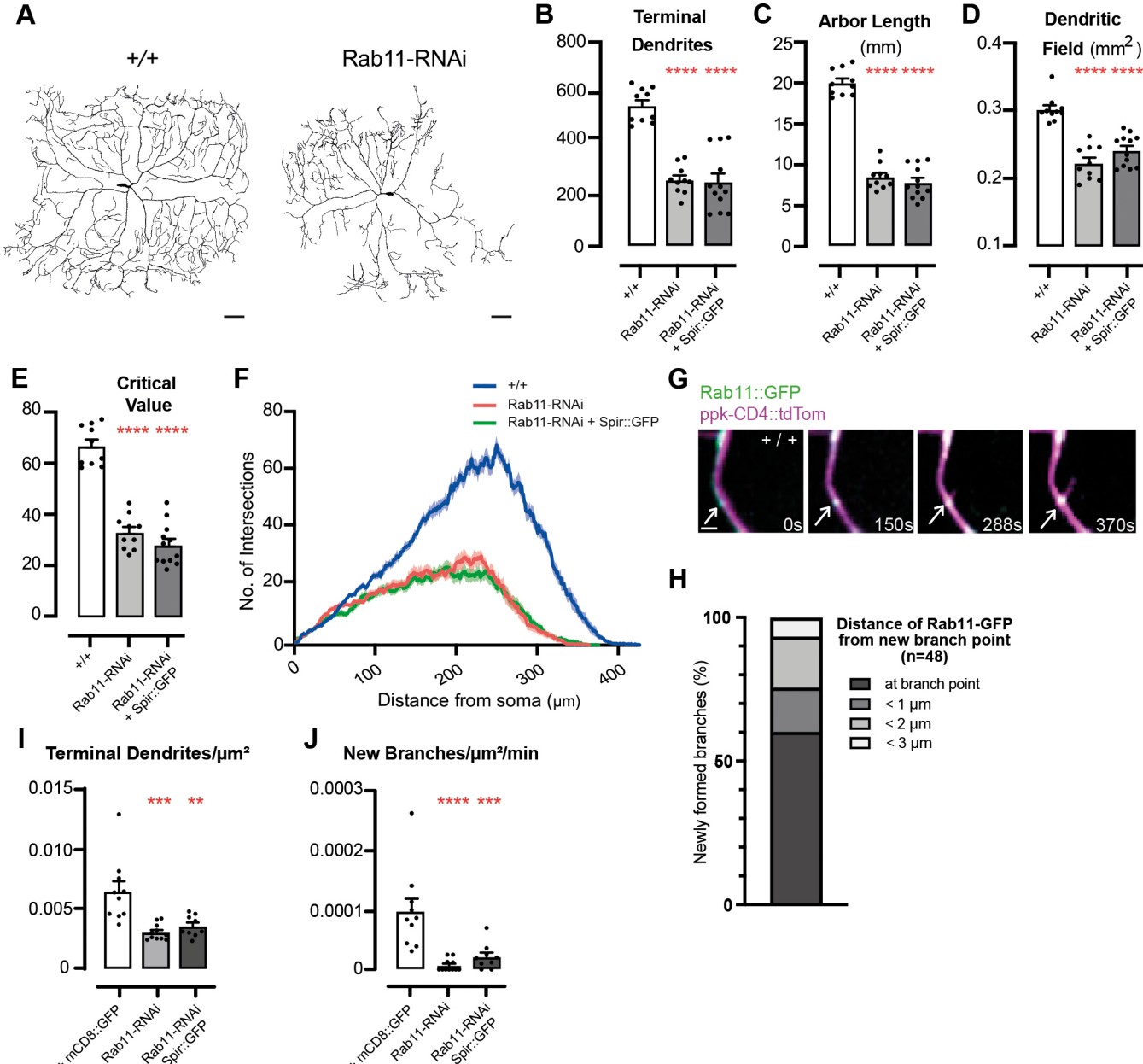

**Fig. 5. Rab11 is required in c4da neurons for dendrite arborization and new branch formation.** (A) Dendrite arbors of c4da neurons in control (+/+) (left) and *ppk-Gal4*-driven Rab11-RNAi (right) L3 larvae. Scale bars: 50 µm. (B-F) Dendrite arbor parameters in control (+/+, *n*=10), *ppk-Gal4*-driven Rab11-RNAi (*n*=10) and *ppk-Gal4*-driven Rab11-RNAi and Spir::GFP (*n*=12). Asterisks indicate significant changes compared to +/+ wild-type controls (white bars) (**$P<0.01$, ***$P<0.001$, ****$P<0.0001$). (B) Total number of terminal dendrites [ANOVA $F_{(2, 29)}=41.06$, $P<0.0001$]. (C) Total length of dendrite arbor (mm) [ANOVA $F_{(2, 29)}=163.6$, $P<0.0001$)]. (D) Dendritic field (mm$^2$) [ANOVA $F_{(2, 29)}=34.00$, $P<0.0001$]. (E) Sholl critical value [ANOVA $F_{(2, 29)}=79.84$, $P<0.0001$]. (F) Sholl profiles. (G) Filmstrip (Movie 7) of control (+/+) L2 larva, where Rab11::GFP (arrows) was observed at the initiation site of a new dendrite branch. Scale bar: 2 µm. (H) Distance of the closest Rab11::GFP puncta from the site of branch initiation (*n*=48) in control (+/+) L2 larvae. (I,J) Dendrite parameters from time-lapse movies of control (+mCD8::GFP, *n*=72 new branches, 10 movies), *ppk-Gal4*-driven Rab11-RNAi (*n*=6 new branches, 11 movies) or *ppk-Gal4*-driven Rab11-RNAi and Spir::GFP (*n*=18 new branches, 9 movies). Asterisks indicate significant changes compared to controls expressing mCD8::GFP (white bars). (I) Terminal branches/µm$^2$ in first movie frame [ANOVA $F_{(2, 27)}=12.41$, $P=0.0002$]. (J) Number of new branches/µm$^2$/min in each movie [ANOVA $F_{(2,27)}=14.29$, $P<0.0001$].

(a positive control), or with Rab6::GFP (a negative control). From lysates of adult heads, Spir::tdTom co-immunoprecipitated with Rab11::GFP and with Capu::GFP, but not with Rab6::GFP (Fig. 6E). This suggests that Spir could selectively associate with Rab11-containing protein complexes in neurons. Consistent with this, we found that, upon simultaneous expression of Spir::GFP and Rab11::mCherry in c4da neurons, Spir::GFP puncta were always accompanied by Rab11::mCherry in dendrites (Fig. 6F-H, 38/38

puncta, *n*=2). Their simultaneous expression did not cause significant changes in the number of terminal branches (Fig. S7A), arbor length (Fig. S7B) or Sholl critical value (Fig. S7D), though dendritic field size was modestly increased (Fig. S7C).

Since Spir and Rab11 colocalized extensively in dendrites, we wondered whether the positioning of one might be dependent on the other. Upon Rab11 knockdown, the punctate distribution of Spir::GFP was unaffected at L3 (Fig. S8A,B) and, in live imaging at L2,

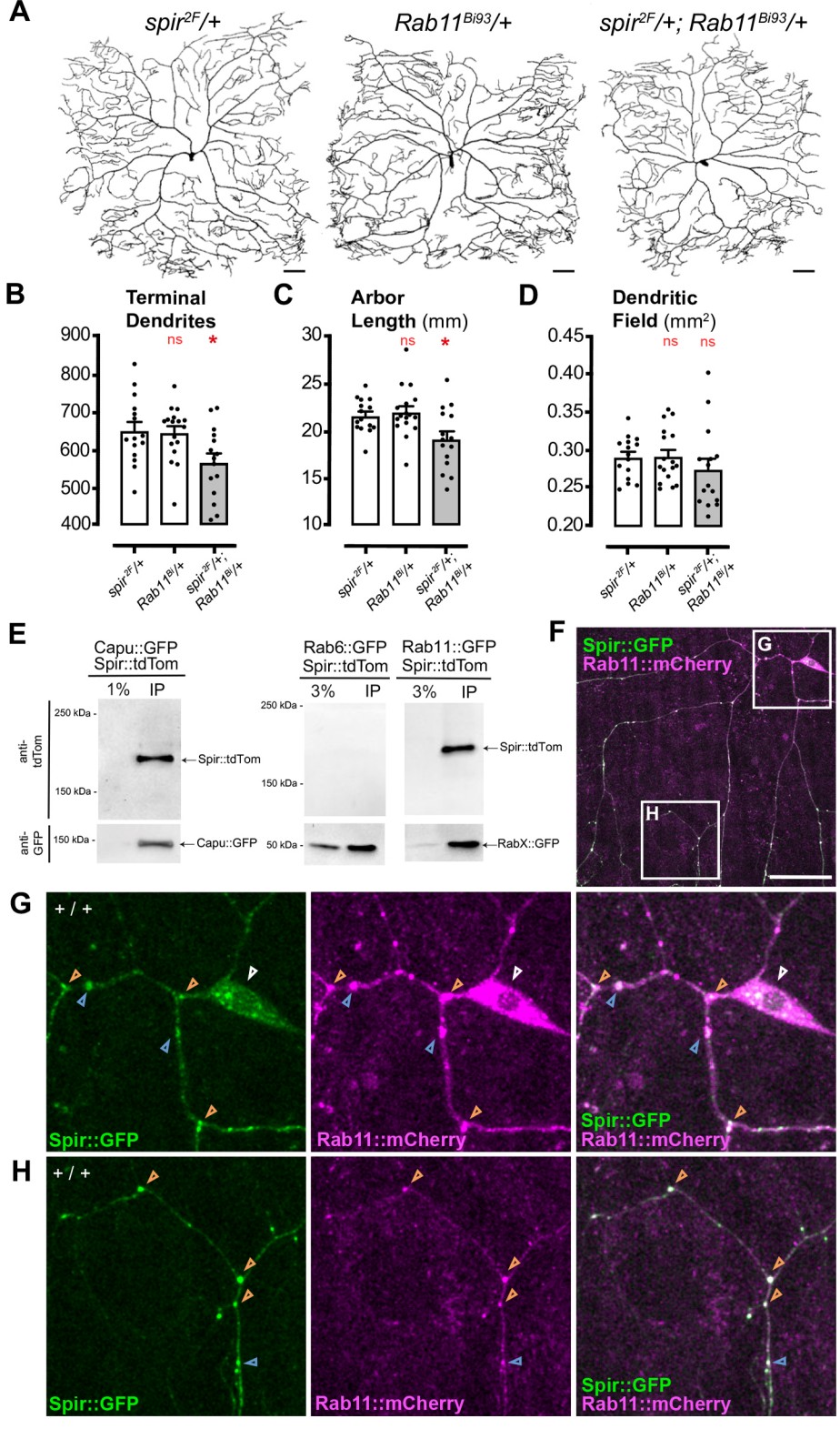

**Fig. 6. Spire interacts with Rab11 in c4da neurons.** (A) Dendrite arbors of c4da neurons in *spir²ᶠ/+* heterozygotes (left), *Rab11^Bi93* heterozygotes (middle) and trans-heterozygotes for both (right). Scale bars: 50 μm. (B-D) Dendrite arbor parameters in *spir* heterozygotes (*spir²ᶠ/+*, *n*=15), *Rab11* heterozygotes (*Rab11^Bi93/+*, *n*=17) and trans-heterozygotes (*spir²ᶠ/+; Rab11^Bi93/+*, *n*=15). Asterisks indicate significant changes compared to *spir²ᶠ/+* heterozygous controls (white bars) (*P<0.05; ns, not significant). (B) Total number of terminal dendrites [ANOVA $F_{(2, 44)}$=4.745, $P$=0.0136]. (C) Total length of dendrite arbor (mm) [ANOVA $F_{(2, 44)}$=5.443, $P$=0.0077]. (D) Dendritic field (mm²) [ANOVA $F_{(2, 44)}$=0.8512, $P$=0.4338]. (E) Spir::tdTom co-immunoprecipitated with either Capu::GFP or Rab11::GFP from adult head lysates, but not with Rab6::GFP. In SDS-PAGE gels, GFP-Trap immunoprecipitates were run alongside 1% or 3% of initial lysate (as indicated), and replicate immunoblots were probed with antibodies to detect GFP or tdTom. (F-H) *ppk-GAL4*-driven Spir::GFP (green) and Rab11::mCherry (magenta). (F-H) Areas outlined in F indicate regions of a c4da neuron cell body selected for magnification (white arrowhead), and proximal dendrites (G) or distal dendrites (H). Colocalization of Spir::GFP and Rab11::mCherry at branch points (e.g. orange arrowheads) and dendrite shafts (e.g. blue arrowheads). Scale bar: 50 μm in F.

Spir::GFP was still positioned at (56%) or near (<2 μm, 44%) each new branch initiation site (*n*=18, Fig. S8C,D, Movie 8). Therefore, Rab11 did not appear to be required for Spir localization at branch initiation sites, and so we wondered if Spir might be required for Rab11 localization instead. Since the Spir-box had potential to bind Rab11, we co-expressed Spir^ΔSB::GFP and Rab11::mCherry, and found that Rab11::mCherry coincided exactly with the abnormal

and more continuous pattern of Spir^ΔSB::GFP in dendrites (Fig. S7E, 35/35 distinct sites, *n*=2), suggesting the ΔSB mutation did not disrupt the association of Spir and Rab11. For technical reasons, we were unable to examine either Rab11::GFP or Rab11::YFP in *spir* mutants, and so we studied the marker Sec15::GFP instead. Sec15 is a Rab11-binding protein (Assaker et al., 2010; Wu et al., 2005; Zhang et al., 2004) and a component of the exocyst, which is a Rab effector

complex of eight proteins involved in the delivery of material to the plasma membrane (TerBush et al., 1996; Wu et al., 2008). Like Rab11::GFP, Sec15::GFP showed punctate, motile expression in dendrite arbors in live imaging at L2, where Sec15::GFP was positioned at (80%) or near (<1 μm, 20%) each initiation site (n=21 new branches in eight movies, Fig. S8E,F, Movie 9). Sec15::GFP was observed at initiation sites in rare instances where nascent branches were captured in *spir* mutants (n=2, from 11 movies, Fig. S8G, Movie 10), suggesting Spir is not required for positioning Sec15::GFP.

We have found that Spir and Rab11 are both required for the synthesis of new dendrite branches and, furthermore, that they genetically interact *in vivo*, physically interact in neurons and colocalize in dendrites. Together these data provide strong evidence that Spir and Rab11 functionally interact with one another in a common molecular pathway for dendrite arborization in c4da neurons.

## DISCUSSION

Spir was identified previously as being regulated by Longitudinals Lacking (Lola), a BTB/POZ transcription factor involved in dendrite development (Ferreira et al., 2014a) and motor axon guidance (Gates et al., 2011). However, the mechanisms of Spir function in neuronal development have remained unclear. Our data presented here indicate that Spir stimulates localized assembly of actin in the seconds prior to outgrowth of new dendrite branches, and that Spir function is linked to the GTPase Rab11 in the formation of new branches and the dynamic motility of existing ones. Unlike Spir::GFP, which arrived from elsewhere in 30% of new branch initiation events, Rab11::GFP was already at the site from the beginning of every movie, suggesting that, at least in some cases, Spir is recruited to a pre-existing site marked by the presence of Rab11. Rab11 has been shown to regulate movement of recycling endosomes toward the plasma membrane (Lasiecka and Winckler, 2011), and the trafficking of receptors and trans-membrane proteins necessary for neurite outgrowth and remodeling (Eva et al., 2010; Park et al., 2004). Sec15 is a co-factor for Rab11 and a subunit of the exocyst complex, which mediates the tethering of secretory vesicles to the plasma membrane prior to fusion and has been shown to be required for c4da dendrite growth, maintenance and regeneration following injury (Peng et al., 2015; Swope et al., 2022). We therefore propose that Spir cooperates with Rab11 to couple actin dynamics with the delivery of lipids and cargoes through the secretory network to the exocyst complex, fueling protrusive outgrowth of nascent dendrites. Consistent with this, Spir function in dendrite arborization required intact domains for actin assembly (KIND and WH2) and for targeting Spir to the secretory network (mFYVE).

The function of Spir in dendrites also required an intact Spir box, which had been suggested to mediate interactions with Rab11. Deleting the Spir-box altered Spir levels and changed its distribution in dendrites from a punctate to a more continuous pattern, but did not disrupt the colocalization of Spir and Rab11::mCherry in dendrites. Additionally, knocking down Rab11 did not affect the localization of Spir::GFP, suggesting that other factors are involved in recruiting Spir to nascent branch initiation sites within arbors. These data raise the possibility that the interaction between Spir and Rab11 is indirect and may be mediated by other unknown factors in protein complexes.

In control animals, newly formed branches were pre-figured either by the appearance of actin polymer *de novo* or by the arrival of pre-formed actin blobs, in roughly equal proportions. In *spir* mutants, there were fewer new branches, and none stemming from initiation sites with *de novo* actin polymer, indicating that Spir is crucial for new dendrite formation from such sites. It is unclear whether

stationary actin patches and motile actin blobs are features of convergent or parallel processes to initiate dendrite branches (Pai and Moore, 2018). Although Spir was present and active at initiation sites stemming from pre-formed actin blobs, it was dispensable for branch initiation from at least some of them. Other molecular mechanisms for branch initiation could involve actin synthesized by the Arp2/3 complex, which has been shown to localize transiently to branch initiation sites in c4da neurons (Stürner et al., 2019). Future experiments are needed to determine whether Spir, Arp2/3 and possibly other actin nucleators function cooperatively, redundantly or in distinct processes for initiating new dendrites.

Spir::GFP overexpression did not affect dendrite arborization (compare Fig. 3 and Fig. S5), supporting the idea that its expression at nascent branch sites reflects that of endogenous Spir. Since Spir::GFP overexpression was not sufficient to induce ectopic branching, and the interval between the stationing of Spir at a branching site and filopodial outgrowth was variable, our findings suggest that Spir activity is tightly regulated and may require additional partners to engage the switch to branch initiation. The KIND domain mutation Y232K is known to abolish the interaction between Spir and the formin Capu, and there is some evidence that Spir and Capu interact to create the spike-like protrusions in dendrites of class 3 da neurons (Stürner et al., 2022). However, in unpublished data, we built a *capu^{GAL4}* line (H.M.B and M.E.Q.) that did not label c4da neurons (D.H. and C.M.), and RNAi lines targeting Capu did not reduce terminal branch number when expressed in c4da neurons (T.F. and D.J.v.M). Of note, the KIND domain also binds intramolecularly to the mFYVE domain (Tittel et al., 2015), and so perhaps the Y232K and C5,6S mutations in these domains disrupt this interaction and thereby impact Spir function in dendrites. Alternatively, Spir might cooperate with another formin in c4da neurons, such as Formin3, which is required for c4da dendrite arborization (Das et al., 2021). Further research will be required to identify a formin partner for Spir in c4da neurons, or to determine if Spir could nucleate actin on its own in neurons, as has been shown *in vitro* (Quinlan et al., 2005).

Interestingly, the Formin3 ortholog INF2 has been reported to interact with mouse Spir-1 in mammalian cells (Manor et al., 2015). Murine orthologs of Spir [Spir-1 (Spire1) and Spir-2 (Spire2)] are expressed in the developing CNS and adult brain (Pleiser et al., 2010; Schumacher et al., 2004). Decreased Spir-2 expression is associated with epilepsy in humans and mice (Hao et al., 2022), but it is not known if either Spir-1 or Spir-2 are involved in dendrite arborization in mammals. A distinct actin nucleator known as cordon-bleu (Cobl) has been demonstrated to promote dendrite branch initiation in cultured hippocampal neurons and slice preparations of cerebellar Purkinje neurons (Hou et al., 2015).

Actin filaments generated by Spir proteins and formins have been shown to provide tracks for myosin-V motors in microtubule-independent transport of Rab11-labeled endosomes to the plasma membrane of oocytes (Holubcová et al., 2013; Schuh, 2011) and in dispersal of Rab27a-labeled melanosomes in melanocytes (Alzahofi et al., 2020). Interestingly, mammalian orthologs of myosin Vb have been shown to bind directly to the orthologs of both Spir and Rab11 (Pylypenko et al., 2016), which together activate myosin Vb and stimulate its motility (Yao et al., 2024). In dendrites, Spir could contribute to an actin pool for myosin-V-mediated delivery of Rab11 vesicles to nascent branches, supplying lipids and proteins required for their protrusive growth. This actin pool could allow the subsequent invasion of microtubules into these nascent branches, perhaps mediated by myosin 6 (Yoong et al., 2020) or augmin (Zhang et al., 2024), and the transition to microtubule-dependent long-

distance transport of cargo-carrying vesicles. While further experiments are required to fully elucidate the precise mechanisms by which Spir, Rab11 and the exocyst promote dendrite branching, our findings show that Spir-mediated actin dynamics link branch initiation with protrusive outgrowth *in vivo*.

## MATERIALS AND METHODS

### *Drosophila* stocks and genetics

Fly stocks were obtained from the Bloomington *Drosophila* Stock Centre [*spir¹*, *spir²ᶠ*, *spir^{DfExel6046}* (referred to as *spir^{Df}*), Mi05646, Mi05737, *UAS-capu::GFP*, *Rab11^{Bi93}*, *w^{1118}*, *UAS-mCD8::GFP*, *UAS-Rab11::GFP*, *UAS-Rab11::YFP*, *UAS-Rab6::GFP*, *UAS-Rab11-dsRNA* (referred to as *UAS-Rab11-RNAi*) and *UAS-LifeAct-GFP*. *ppk-GAL4*, *ppkCD4::tdTom* and *ppkCD4::GFP* were gifts from Dr Yuh-Nung Jan (University of California, San Francisco, CA, USA). *UAS-Rab11::mCherry* and *UAS-sec15-GFP* were gifts from Dr Gregory Emery (Université de Montréal, Canada). The *nSyb-GAL4* line was created by Dr Julie Simpson (University of California, Santa Barbara, CA. USA), for which we used 2nd chromosome insertions generated by Dr Stefan Thor (University of Queensland, Australia). To create the lines *UAS-spir::GFP*, *UAS-spir::tdTom*, *UAS-spir^{Y232K}::GFP*, *UAS-spir^{ABCD}::GFP*, *UAS-spir^{ASB}::GFP* and *UAS-spir^{C5,6S}::GFP*, DNA constructs encoding full-length Spir or each of the four mutated versions (Kerkhoff et al., 2001; Quinlan, 2013; Quinlan et al., 2005) were designed to add C-terminal GFP or tdTomato (tdTom) fluorophores. These constructs were subcloned into the vector pMUH, then microinjected using standard procedures (Bestgene) to target the VK00027 landing site in the *Drosophila* genome with φc31-mediated integration (Pfeiffer et al., 2010). *spir^{GAL4}* was generated by inserting a so-called Trojan-GAL4 into the *spir* locus (Diao et al., 2015; Venken et al., 2011). pT-GEM(0), a gift from Benjamin White (Addgene plasmid #62891) was microinjected (Bestgene) into the MiMIC line Mi05646 (Diao et al., 2015; Venken et al., 2011). The same plasmid was injected into the MiMIC line Mi05737 to generate *capu^{GAL4}*. CRISPR/Cas9-mediated homologous recombination was used to introduce spaghetti-monster GFP-HA (smGFP-HA) (Viswanathan et al., 2015) into the *spir* locus. To generate Spir-smGFP-HA flies, we used an approach combining dual guide RNA (gRNA) sequences (Bence et al., 2017) and short homology arms (Kanca et al., 2019). The CRISPR Optimal Target finder (http://targetfinder.flycrispr.neuro.brown.edu/index.php) was used to select gRNA sequences for the C-terminus of Spir (5′-GTCGGCCCTGGATCTGACGCCCGTC-3′) and immediately after the 3′UTR (5′-GTCGGCAAACTAAAGAACAAGATTC-3′), and these oligonucleotides were cloned into pCFD3-dU6:3gRNA (Addgene plasmid #49410) (Port et al., 2014). Homology sequences on 200 nucleotides (PAM removed) were cloned into a self-linearizable Puc573 vector for homologous recombination (Kanca et al., 2019). The repair template included smGFP-HA, the endogenous 3′UTR of Spir and the fluorescent eye reporter 3xP3-dsRed, all flanked by two PiggyBac recombinase sites. Plasmids expressing the gRNAs (100 ng/µl) and donor template (250 ng/µl) were mixed and micro-injected into nos-Cas95 embryos (Ren et al., 2013) (BDSC 78782). Progeny were screened via fluorescence for the 3xP3-dsRed reporter, and fly stocks were established. All injections and initial screening were completed by BestGene. Proper insertion of smGFP was confirmed via genomic PCR (primers 5′-GGGGATTCAACCTGTTCTCCT-3′ and 5′-TGTGCAAGTGCGTTCTG-AAG-3′), and expression confirmed by western blot.

### Antibodies

To produce anti-Spir antisera, DNA encoding Spir amino acids 565-692 were cloned into pGEX-6P-1 to produce an IPTG-inducible C-terminal GST fusion proteins in BL21 cells. The purified GST fusion protein was injected into two rabbits and the serum was extracted according to approved protocols at the Comparative Medicine and Animal Resources Centre (McGill University) and to standards established by the Canadian Council on Animal Care. Crude serum (500 µl) was then pre-absorbed by overnight incubation with 500 µl L3 larvae (*spir¹/spir^{Df}*) that had been fixed previously in 4% PFA. Other primary antibodies for immunohistochemistry were Alexa-647-conjugated goat anti-HRP (1:800, Jackson ImmunoResearch, #123605021) and rabbit anti-HA (1:300, Cell Signaling Technology, #3724), while the secondary antibody for

anti-Spir was Alexa-647-conjugated goat anti-rabbit (1:1000, Invitrogen, #A21244), and for anti-HA immunohistochemistry it was Alexa-405-conjugated goat anti-rabbit (1:1000, Invitrogen, #A48254). For western blotting, membranes were incubated with mouse anti-GFP (1:10,000, Clontech, #632380), rabbit anti-DsRed (1:500, Takara, #632496, for tdTom detection), mouse anti-Actin (1:5000, Sigma, #A4700) or rabbit anti-Spire serum (directed at amino acids 565-692; 1:1000, pre-absorbed, as described above). For detection, HRP-conjugated secondary antibodies [anti-mouse and anti-rabbit (both 1:3000, Bio-Rad, #1706516 and #1706515, respectively)] were used and revealed with chemiluminescence (Amersham ECL Western Blotting Detection Reagents).

### Immunohistochemistry

L3 larvae of either sex were dissected in Sorensen phosphate buffer (pH 7.4) and fixed in 4% paraformaldehyde (in Sorensen's buffer) for 10 min at room temperature. Three washes over 20 min were carried out with Sorensen's buffer containing 0.2% Triton-X-100 (Sigma). Specimens were then blocked with Sorensen's buffer containing 5% normal goat serum (NGS), incubated at 4°C with primary antibodies overnight, washed as above, then incubated with secondary antibodies for 2 h at room temperature. Anti-Spir antiserum for immunohistochemistry was directed at amino acids 1-91. Primary and secondary antibodies were diluted in Sorensen's buffer containing 5% NGS. After a final wash as above, specimens were then mounted with SlowFade Diamond Antifade Mountant (Invitrogen) for imaging.

### Co-immunoprecipitation and western blotting

Lysates for co-immunoprecipitations and western blots were made from adult heads aged 1-5 days after eclosion. For western blots, heads were mechanically crushed using a motorized pestle directly in 2× Laemmli buffer. For co-immunoprecipitations, 100 heads were cut and collected on dry ice, then mechanically crushed using a motorized pestle in 100 µl of lysis buffer containing 50 mM Tris (pH 8.0), 150 mM KCl, 1 mM MgCl₂, 1 mM EGTA, Roche protease inhibitor cocktail and 10% glycerol. Lysates were treated with 10% deoxycholic acid and 10% Triton X-100, then incubated overnight on a Nutator with anti-GFP-bound agarose beads (Chromotek GFP-Trap #gta). Lysates and beads were separated in a centrifuge, washed with a glycerol-free buffer, and the complexes on the beads were dissociated directly in Laemmli buffer, then separated using standard SDS-PAGE and wet-transfer procedures.

### Imaging

All imaging and quantification was of ddaC, the dorsal-most c4da neuron, in abdominal segments of the larval body wall. For static images, L3 larvae were dissected and prepared for immunohistochemistry, or they were squished directly on the slide in 100% glycerol and prepared for imaging of native fluorescence (GFP, tdTom, mCherry or mRFP) within 10 min. Static images were taken with an Olympus Fluoview FV-1000/BX63 confocal microscope using either a 20× or 60× oil-immersion objective lens. All time-lapse imaging was carried out on L2 larvae, where eggs were initially collected for 4 h on agar plates containing grape juice and yeast. After 24 h, hatched larvae of selected genotypes were transferred on a fresh fly food plate to start timing.

L2 larvae (65 h±4 h AEL) were mounted in a drop of halocarbon oil in a polydimethylsiloxane (PDMS) microfluid device, and then imaged by live confocal microscopy (Zeiss, Spinning Disk, Inverted Axio Observer Z1) with 63× objective lens, at room temperature with Immersol W (Zeiss) between lens and coverslip. Using Zen software (Zeiss), live imaging of fluorescent proteins GFP and tdTomato was performed for 5 min with 5 s intervals, on average. If only GFP was being imaged, then the interval was 2 s, on average. Only movies obtained from larvae that were still alive and active after imaging were used for analysis. Each movie was then registered on ImageJ using StackReg plugin before analysis (Thevenaz et al., 1998).

### Quantification of Lifeact::GFP and Spir::GFP localization

New branches appearing in each movie were identified, and initiation sites were classified as having *de novo* actin polymer if a punctum of LifeAct::GFP intensity appeared at the site and grew in intensity until protrusion of the new branch. Initiation sites had 'pre-formed' actin if motile blobs of

LifeAct::GFP were observed to arrive prior to branch outgrowth (motile). Instances where a new branch grew from a site where a LifeAct::GFP punctum existed in the first frame of the movie were disregarded because the movie did not capture the origin of the actin punctum. Consistent with a previous report (Nithianandam and Chien, 2018), we observed only a single rare instance where a filopodium grew from an initiation site where LifeAct:: GFP was not detected. Overall LifeAct::GFP expression in *spir* mutants appeared lower than in controls, consistent with our previous report for Spir knockdown with RNAi (Ferreira et al., 2014a), but it was certainly sufficient to monitor actin polymer at sites of branch initiation.

To measure the association of Spir::GFP labeled punctae to either a dendrite shaft, branch point or tip, a custom routine was developed to extract this information from images acquired with a 20× objective (two to four images stitched together where necessary using the Pairwise Stitching plugin (Preibisch et al., 2009). From maximum intensity projections (MIPs) of 10 c4da neurons (ddaC) labeled for the dendrite marker CD4-tdTom, signal from beyond the cell of interest was cleared manually, the background was subtracted, and a MIP mask created. This mask was then applied to the SpirFL::GFP channel. Pixels outside the mask were cleared. The MIP mask was skeletonized using Fiji's skeletonization plugin (Arganda-Carreras et al., 2010) to obtain the location of shafts, branch points and tips. A script from the Neuroantomy update site was then used to determine which Spir:: GFP punctae were associated with such skeleton features. When a Spir::GFP punctum was 3 µm or less (snap-to distance) from either a branch point or tip, it was assigned to each, respectively. This snap-to distance was selected because, from a representative Spir::GFP image, 80% of the punctae had a diameter of 3 µm or less. Additionally, no branch points or tips were 3 µm or less apart from one another. The number of punctae at branch points and tips was then divided by the total number of branch points and tips, respectively, to obtain the percentage of branch points and tips that were occupied by Spir::GFP.

### Dendrite morphometry

For morphometry of c4da neurons (ddaC) at L3, images were acquired at 20×, with one image field usually sufficient to capture the entire dendritic field of a neuron. However, in some cases two images were stitched together using the Fiji plugin Pairwise Stitching to generate an image of the entire neuron (Preibisch et al., 2009). Maximum intensity projections of confocal *z*-series image stacks were segmented as described previously (Ferreira et al., 2014a). Various Fiji plugins were used to determine branching characteristics of neurons: Sholl Analysis to generate Sholl profiles and extract Sholl-related parameters (Ferreira et al., 2014b), Strahler Analysis to assess the number of terminal dendrites (Ferreira et al., 2014b; Strahler, 1957), and Analyze Skeleton for total arbor length. The dendritic field was determined by fitting a polygon around the segmented neuron on Fiji, and measuring the area inside the shape (Schindelin et al., 2012). For morphometry of ddaC neurons at L2 in time-lapse movies, the number of terminal dendrites/µm$^2$ was calculated in the first frame of each movie. For the duration of each movie, the number of new branches/µm$^2$/minute was calculated.

### Measuring dendrite dynamics

Pre-existing dendrites at the beginning of live-imaging movies were analyzed for the dynamics of their terminal branches using MTrackJ (Meijering et al., 2012). Terminal branch length was compared by an observer for every frame of each movie. Dendrites that extended or retracted, or did both, or remained static were recorded, and these numbers were expressed as the percentage of the total number of pre-existing dendrites at the beginning of each movie. Newly formed dendrites were tracked similarly, and their protrusive outgrowth was measured, where mean speed was the average of the speeds of growth for each frame (including extensions and pauses) until the branch stopped growing altogether, or retracted, or the movie ended. The proportion of time the branch paused for during this period of saltatory outgrowth was calculated, as was the maximum length of its protrusion.

### Graphs and statistics

Bar graphs and statistics were created and analyzed using GraphPad Prism v10, and displayed data are expressed as mean±standard error (s.e.m.), with dots showing individual data points. For comparisons of more than two conditions, data were tested for normal distribution using D'Agostino & Pearson test, and analyzed for statistical significance using one-way ANOVA (multiple groups) and then Dunnett's post-hoc tests. For the repeated measures of western blots, the Friedman test was applied then Dunn's post-hoc tests. To compare two conditions, Student's *t*-test (unpaired, two-tailed) was used. Asterisks in graphs indicate the significance of *P*-values compared to the indicated control group, which is the left-most white bar in each bar graph (*$P<0.05$, **$P<0.01$, ***$P<0.001$, ****$P<0.0001$; ns, not significant).

### Acknowledgements
The authors thank Greg Emery, Peter McPherson and Martine Girard for advice and sharing reagents, and their colleagues Emilie Peco, Eunjoo Cho, Adela Ralbovska, Keith Murai, Sabrina Chierzi and Todd Farmer for helpful suggestions. For technical support, the authors thank Dr Min Fu and Shi-bo Feng of the RI-MUHC Molecular Imaging Platform, and Tim McLean. Stocks obtained from the Bloomington *Drosophila* Stock Center (NIH P40OD018537) were used in this study.

### Competing interests
The authors declare no competing or financial interests.

### Author contributions
Conceptualization: M.E.Q., D.J.v.M.; Data curation: D.H., C.M., C.H., T.F., L.C.-M., Y.O., D.J.v.M.; Formal analysis: D.H., C.M., C.H., T.F., L.C.-M., Y.O., H.M.B., D.J.v.M.; Funding acquisition: M.E.Q., D.J.v.M.; Investigation: D.H., C.M., C.H., T.F., Y.O., W.S., H.M.B.; Methodology: D.H., C.M., C.H., T.F., Y.O., W.S., H.M.B.; Project administration: D.J.v.M.; Software: T.F.; Supervision: M.E.Q., D.J.v.M.; Validation: D.H., C.M., C.H., T.F., L.C.-M., Y.O., W.S., H.M.B., D.J.v.M.; Visualization: D.H., C.M., C.H., D.J.v.M.; Writing – original draft: D.H., C.M., C.H., M.E.Q., D.J.v.M.; Writing – review & editing: C.M., D.J.v.M.

### Funding
This work was supported by the Canadian Institutes of Health Research (FRN137034 to D.J.v.M.), the National Institutes of Health (R01GM096133 to M.E.Q.), a Ruth L. Kirschstein National Research Service Award (GM007185 to H.M.B.), and studentships from the Research Institute of the McGill University Health Centre and McGill University (to D.H.). Open Access funding provided by McGill University. Deposited in PMC for immediate release.

### Data and resource availability
All relevant data and details of resources can be found within the article and its supplementary information.

### Peer review history
The peer review history is available online at https://journals.biologists.com/dev/lookup/doi/10.1242/dev.204786.reviewer-comments.pdf.

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
