## [Peer Review File · Development (Cambridge, England)]

Nascent dendrite branches initiated by a localized burst of Spire-dependent actin polymerization.

Deirdre Hatton, Claire Marquilly, Caitlin Hanrahan, Tiago Ferreira, Yimiao Ou, Lauren Cinq-Mars, Will Silkworth, Hannah M. Bailey, Margot E. Quinlan and Donald J. van Meyel
DOI: 10.1242/dev.204786

Editor: Thomas Lecuit

Review timeline

Original submission:	11 March 2025
Editorial decision:	2 May 2025
First revision received:	14 July 2025
Accepted:	4 September 2025

Original submission

First decision letter

MS ID#: dev.204786

MS TITLE: Nascent dendrite branches initiated by a localized burst of Spire-dependent actin polymerization.

AUTHORS: Deirdre Hatton, Claire Marquilly, Caitlin Hanrahan, Tiago Ferreira, Lauren Cinq-Mars, Yimiao Ou, Will Silkworth, Hannah M. Bailey, Margot E. Quinlan and Don van Meyel

Dear Dr van Meyel,

I have now received all the referees reports on the above manuscript, and have reached a decision. The referees' comments are appended below, or you can access them online: please go to .

The overall evaluation is positive and we would like to publish a revised manuscript in Development, provided that the referees' comments can be satisfactorily addressed. Please attend to all of the reviewers' comments in your revised manuscript and detail them in your point-by-point response. If you do not agree with any of their criticisms or suggestions explain clearly why this is so. If it would be helpful, you are welcome to contact us to discuss your revision in greater detail. Please send us a point-by-point response indicating your plans for addressing the referees' comments, and we will look over this and provide further guidance.

Reviewer 1

Advance summary and potential significance to field

Summary of the advance made in this paper and its potential significance to the field. This study by Hatton, Marquilly and colleagues reports on the role of Spire, an actin nucleating factor, in dendrite branch formation and dynamics using the da neuron model in *Drosophila* larvae. It also investigates genetic interactions between Spire and Rab11-dependent endocytic trafficking in these aspects of dendritogenesis.

The study is elegant, as it combines time-lapse imaging of dendrite dynamics and phenotypic analyses of Spir and Rab11 mutants, and provides detailed quantifications. The main conclusions are that Spir promotes actin polymerization at nascent branch points, leading to new branch formation, and that this pathway cooperates with Rab11-mediated secretory trafficking to establish the highly-branched arbor structure of Type IV da neurons. These conclusions are supported by high-quality data and clear reporting of the findings. In particular, the comprehensive analyses of Spir and Rab11 in dendritogenesis - including time lapse imaging of dendrite dynamics and Lifeact+ actin polymerization, as well as their protein localization- are the major strengths of the paper.

Overall, the manuscript presents an important advance linking Spir's cytoskeletal function to dendrite branch initiation and growth *in vivo*. It also denotes the importance of multiple Spir domains in this process, and the intersection with Rab11-dependent endocytic recycling, which provides new information for further mechanistic studies. I found the experimental designs and genetic manipulations to be rigorous, and the quantifications rather comprehensive. The manuscript is also well-written.

Comments for the author

- The manuscript abruptly shifts from analyses of Spir domain mutants to demonstration of physical interactions between Spir and Rab11 by colP from *Drosophila* extracts. Thus, there is a missed opportunity to test whether the interaction is dependent on the Spir domain, and via the mFYVE in particular, given the stated relationship of this domain with the secretory network. The discussion dismisses this possibility based on the lack of 'interdependent localization' (line 291+). However, this point strikes me as premature, as the interaction could be directly/indirectly mediated via mFYVE and PIP membrane organization? Testing the Spir deletion mutants with Rab11 in colPs could further strengthen the structure-functional aspects of this study, and more strongly integrate the two in this model. At the very least, the authors should elaborate in the discussion on the reasoning for discounting involvement of Spir domains driving these interactions with Rab11.

Minor comments re. the text:

-The figure legends describe the ANOVA tests but do not describe the pairwise comparisons that are significantly different, and what is being compared. For instance in Fig 3C-F, are the phenotypes with asterisks compared to the WT? Is there a statistical difference between the Spir-GFP rescue compared to the mutants?

-For Figure legend 4A, specify that the white box denotes the spir2F/spirDf group.

Reviewer 2

Advance summary and potential significance to field

Actin polymerization plays an important role in dendrite branch initiation and growth. Hatton et al. studied the role of Spire, a multidomain protein that acts as an actin nucleating factor, in *Drosophila* sensory c4da neurons. They report that spire drives local actin nucleation at points of branch initiation during dendrite development. Local actin nucleation visualized by LifeAct, which formed stationary puncta before branch initiation, were dependent on spire. Similarly, Spire localized to specific sites before branch initiation. The authors further explored the role of Rab11 based on the interaction of Spire-Rab11 in other tissues and showed a similar and cooperative action of both factors, while neither Rab11 nor Spire could compensate for the loss of the other. They further show physical interaction and colocalization between Spire and Rab11 suggesting that they form a complex during branch initiation. However, the presence of Spire or Rab11 was not required for punctate localization at branch initiation points of the other interaction partner (Sec15GFP was used instead of Rab11::GFP).

Overall, the authors' work suggests cooperative functions of actin nucleators like spire and Rab11-positive recycling endosomes are required for dendrite branch initiation and dynamics during dendrite development. The work is quite thorough overall and provides novel insights into the function of Spire/Rab11 in a suitable *in vivo* system. At the same time, it currently provides only limited mechanistic insight into how spire acts together with Rab11 structures to promote branch

initiation, and how this might be linked to known branch initiation effectors like Arp2/3. Nonetheless, the study has merit and provides novel insight into the complexity of dendrite branch initiation and the control of actin nucleation.

Comments for the author

I have a few requests that should be addressed to tie in some loose ends and for thoroughness:

1. Spire is expressed in different body wall cells including other DA neurons and epithelial cells. The authors mostly use whole mutant animals, how do they know their findings are due to cell autonomous function of spire? While spire-GFP largely rescues branching in the mutant background, it does not rescue arbor length and field size. How does the spire-RNAi phenotype compare to the whole mutant?
2. Does spire-GFP overexpression affect dendrite morphology or branching? While it rescues global loss of spire to a large extent, it might affect dynamics in the wildtype background. This has not been shown, also not during comparison of the different overexpressed Spire deletion constructs.
3. Similar to Spire, Arp3 has been shown to localize at branch initiation sites before outgrowth (Sturner et al., 2019). Can the authors explore or at least comment on the relationship between spire and arp2/3 dependent branch initiation? Are they acting in sync or partially redundant, or actually involved in distinct initiation processes? It would have been possible to covisualize arp3 and Rab11 or Spire to address this point based on the existing and published arp3-GFP transgene and the similarity of arpc1 and Spire mutant phenotypes.
4. When looking for Spire/Rab11 colocalization, some quantitative measure is necessary as the authors just show exemplary images of colocalized Spire/Rab11 puncta. Are all or most puncta colocalized independent of their position on the dendritic arbor?
5. The authors did a nice domain analysis of Spire and showed that Rab11 and Spire are physically associated, but not if the putative SPIR-box interaction domain or other Spire domains are relevant in this case. It would be of interest to look at colocalization or biochemical interaction of Rab11 with the Spire deletion transgenes. It should be feasible to test if, e.g., Spire Δ SB is still associated with Rab11. This should at least provide some insight about the physical association of these two effectors.
6. Did the authors notice changes in dendrite morphology upon co-overexpression of Spire and Rab11? Since they show some degree of cooperativity, this might be different from individual overexpression. Alternatively, as they show a dosage sensitive cooperativity in a trans-heterozygous background, testing a reciprocal "rescue" under these conditions might be revealing.

Minor:

1. Please show bar graphs with individual data point, which makes the data more transparent and is a better visual guide than error bars only.
2. Type of error bars should be mentioned in all figure legends, I guess SEM as stated in the statistics part?
3. In their discussion the authors mention that Spire could contribute to an actin pool for myoV-dependent delivery of Rab11 vesicles to nascent branches, yet Sec15 localization is not affected in Spire mutants. Did the authors observe entry of Rab11 into nascent branches in their controls, which seems to be the case for Sec15 (Fig. S7G)?

First revision

Author response to reviewers' comments

In response to the reviewers' Suggestions to Authors (in black below), we have conducted additional experiments and made revisions as detailed in our responses (in blue). We believe these additional experiments and revisions have strengthened the manuscript considerably.

SUGGESTIONS TO AUTHORS

Reviewer 1

The manuscript abruptly shifts from analyses of Spir domain mutants to demonstration of physical interactions between Spir and Rab11 by colP from *Drosophila* extracts. Thus, there is a missed opportunity to test whether the interaction is dependent on the Spir domain, and via the mFYVE in particular, given the stated relationship of this domain with the secretory network. The discussion dismisses this possibility based on the lack of 'interdependent localization' (line 291+). However, this point strikes me as premature, as the interaction could be directly/indirectly mediated via mFYVE and PIP membrane organization? Testing the Spir deletion mutants with Rab11 in colPs could further strengthen the structure-functional aspects of this study, and more strongly integrate the two in this model. At the very least, the authors should elaborate in the discussion on the reasoning for discounting involvement of Spir domains driving these interactions with Rab11.

Response: We appreciate the opportunity to clarify this point, which has been addressed in the revised manuscript with a new experiment (Fig. S7E, line 258), and with text revisions in the discussion (line 300). By “interdependent localization”, we simply meant that Rab11 knockdown did not affect localization of Spir::GFP, and Spir knockdown did not affect localization of Sec15-GFP, a Rab11 effector and surrogate for Rab11 localization. Although RNAi knockdowns might not deplete all the Rab11 or Spir from c4da neurons in these experiments, this data suggests that the interaction between Spir and Rab11 seen in co-immunoprecipitation is indirect and perhaps mediated by other factors in protein complexes.

As noted (line 62), the Spir-box had potential to mediate interactions with Rab11. Deleting the Spir-box altered Spir levels and changed its distribution in dendrites from a punctate to a more continuous pattern. In a newly added experiment (Fig. S7E, line 258), we co-expressed Spir Δ SB::GFP and Rab11::mCherry and found that Rab11::mCherry coincided exactly with the abnormal and more continuous pattern of Spir Δ SB::GFP (Fig. S7E). Since deletion of the Spir-box did not disrupt the co-localization of Spir and Rab11::mCherry in dendrites, we did not pursue whether the Δ SB mutation could disrupt the co-IP of Spir and Rab11.

The reviewer asks if the interaction could be mediated by the mFYVE domain, but as noted (line 178), FYVE domains interact with lipids, particularly PtdIns3P and so any effect of mutating the mFYVE domain (the C5,6S mutation) on the interaction with Rab11 is likely to be indirect. Furthermore, the C5,6S mutation did not affect the punctate distribution of Spir::GFP in dendrites (Fig. S4C, and line 188), suggesting it does not affect endomembrane targeting or trafficking as might be expected if it disrupted interaction with Rab11. We note (line 331) the possibility that the C5,6S mutations might disrupt intramolecular interaction with the KIND domain instead.

Minor comments re. the text:

-The figure legends describe the ANOVA tests but do not describe the pairwise comparisons that are significantly different, and what is being compared. For instance in Fig 3C-F, are the phenotypes with asterisks compared to the WT? Is there a statistical difference between the Spir-GFP rescue compared to the mutants?

Response: This is now clarified in the Materials and Methods section (line 511), and in each Figure legend. We also added notation (“ns”) to every bar graph where the experiment group was not significantly different from the control.

-For Figure legend 4A, specify that the white box denotes the spir2F/spirDf group.

Response: This is done, and we clarified that the controls are shown as white bars in all similar bar graphs (line 511).

Reviewer 2

1. Spire is expressed in different body wall cells including other da neurons and epithelial cells. The authors mostly use whole mutant animals, how do they know their findings are due to cell autonomous function of spire? While spire-GFP largely rescues branching in the mutant background, it does not rescue arbor length and field size. How does the spire-RNAi phenotype compare to the whole mutant?

Response: Importantly, this paper focuses on the role that Spir plays in new branch formation to establish the correct number of terminal branches. Terminal branch number is rescued by expression of Spir in c4da neurons of *spir* mutants, and therefore appears to be cell

autonomous. As noted (line 55), we previously showed that knockdown of *Spir* with RNA interference (RNAi) in *cd4da* neurons reduced the number of terminal dendrite branches and total arbor length (Ferreira et al. 2014), consistent with the phenotype we report here for null mutants (Fig. 3). Since total arbor length is reduced by *Spir* RNAi in *c4da* neurons but is not fully rescued by expression of *Spir* in *c4da* neurons of *spir* mutants, our findings are consistent with possibility that (line 162) “perhaps additional cell types contribute to total arbor length, such as epithelial cells in the larval body wall where *Spir* is expressed”.

2. Does *spire*-GFP overexpression affect dendrite morphology or branching? While it rescues global loss of *spire* to a large extent, it might affect dynamics in the wildtype background. This has not been shown, also not during comparison of the different overexpressed *Spire* deletion constructs.

Response: As noted (line 319), “*Spir::GFP* overexpression did not affect dendrite arborization (compare Fig. 3 and Fig. S5).” The results for controls (Fig 3, *spir*^{2F/+} heterozygotes or *spir*^{Df/+} hemizygotes, white bars) and *Spir* overexpression (Fig. S5, +*Spir* (+/+), white bar), are nearly identical. These results were gathered in the context of distinct experiments and so, for rigor and clarity, they were not plotted on the same graph.

3. Similar to *Spire*, *Arp3* has been shown to localize at branch initiation sites before outgrowth (Sturner et al., 2019). Can the authors explore or at least comment on the relationship between *spire* and *arp2/3* dependent branch initiation? Are they acting in sync or partially redundant, or actually involved in distinct initiation processes? It would have been possible to covisualize *arp3* and *Rab11* or *Spire* to address this point based on the existing and published *arp3*-GFP transgene and the similarity of *arpc1* and *Spire* mutant phenotypes.

Response: In the future it will be fascinating to dissect the relationship between different actin nucleators like *Spire* and *Arp2/3* and branch initiation. Sturner et al (2019) showed *Arp3* is localized at branch initiation sites before outgrowth, but they did not demonstrate that *Arp3* activity regulates F-actin there, as we have done for *Spir* in this manuscript.

Therefore, at this time we cannot be sure if *Arp3* has similar F-actin nucleating activity and dynamics as *Spir* at branch initiation. Perhaps this can be addressed in a future manuscript, along with the co-visualization experiments proposed by the reviewer.

However, the current *Spire*-mCherry-based tools for co-visualization with *Arp3*-GFP are suboptimal because of their relatively low fluorescence signal, which does not permit in live imaging of new branches. Also, because they are mCherry based, they cannot be seen in the context of the marker we most often use for dendrites (*tdTomato*). Reporters with new, alternative fluors need to be generated and validated, which is beyond the scope of this manuscript.

In the discussion, we did address this point (line 314), but we have now added the following (line 318): “Future experiments are needed to determine whether *Spir*, *Arp2/3*, and possibly other actin nucleators function cooperatively, redundantly, or in distinct processes for initiating new dendrites”.

4. When looking for *Spire*/*Rab11* colocalization, some quantitative measure is necessary as the authors just show exemplary images of colocalized *Spir*/*Rab11* puncta. Are all or most puncta colocalized independent of their position on the dendritic arbor?

Response: Quantification of this has been added (line 247), and the figure has been improved for clarity (Fig 6F-H). Within *c4da* dendrite arbors, all *Spir::GFP* puncta had *Rab11::mCherry*. Within and near *c4da* cell bodies, *Rab11::GFP* was more widespread.

5. The authors did a nice domain analysis of *Spire* and showed that *Rab11* and *Spire* are physically associated, but not if the putative SPIR-box interaction domain or other *Spire* domains are relevant in this case. It would be of interest to look at colocalization or biochemical interaction of *Rab11* with the *Spire* deletion transgenes. It should be feasible to test if, e.g., *Spire* Δ SB is still associated with *Rab11*. This should at least provide some insight about the physical association of these two effectors.

Response: Please see our first response to Reviewer 1 above, where describe a new experiment (Fig. S7E in revised manuscript, line 258) in which we co-expressed *Spir* Δ SB::GFP and *Rab11::mCherry* and found that *Rab11::mCherry* coincided exactly with the abnormal and

more continuous pattern of *Spir* Δ SB::GFP (Fig. S7E). We clarified this in the discussion (line 301).

6. Did the authors notice changes in dendrite morphology upon co-overexpression of Spire and Rab11? Since they show some degree of cooperativity, this might be different from individual overexpression. Alternatively, as they show a dosage sensitive cooperativity in a trans-heterozygous background, testing a reciprocal "rescue" under these conditions might be revealing.

Response: We did this experiment and provide the results in a new supplementary figure (Fig. S7 in revised manuscript). Co-expression of *Spir*::GFP and *Rab11*::mCherry in *c4da* neurons did not cause significant changes in the number of terminal branches (Fig. S7A), arbor length (Fig. S7B), or Sholl critical value (Fig. S7D), though dendritic field was modestly increased (Fig. S7C). We did not attempt to rescue trans-heterozygotes, but in somewhat related experiments, we noted that expressing *Spir*::GFP did not rescue the effects of *Rab11* RNAi (Fig. 5B-F, I-J), nor did *Rab11*:YFP rescue *spir*^{2F/Df} mutants.

Minor:

1. Please show bar graphs with individual data point, which makes the data more transparent and is a better visual guide than error bars only.

Response: This has been done with all bar graphs, and noted in line 505

2. Type of error bars should be mentioned in all figure legends, I guess SEM as stated in the statistics part?

Response: This has been done with all bar graphs. Yes, it is SEM.

3. In their discussion the authors mention that Spire could contribute to an actin pool for myoV-dependent delivery of Rab11 vesicles to nascent branches, yet Sec15 localization is not affected in Spire mutants. Did the authors observe entry of Rab11 into nascent branches in their controls, which seems to be the case for Sec15 (Fig. S7G)?

Response: No, we did not see large Rab11 puncta enter nascent branches in our controls, though it is possible that individual Rab11 vesicles are too small and faint to resolve in live imaging of dendrites. In the discussion (line 357), we write "While further experiments are required to fully elucidate the precise mechanisms by which Spire, Rab11 and the exocyst promote dendrite branching, our findings show that Spire-mediated actin dynamics link branch initiation with protrusive outgrowth in vivo."

Second decision letter

MS ID#: dev.204786R1

MS TITLE: Nascent dendrite branches initiated by a localized burst of Spire-dependent actin polymerization.

AUTHORS: Deirdre Hatton, Claire Marquilly, Caitlin Hanrahan, Tiago Ferreira, Lauren Cinq-Mars, Yimiao Ou, Will Silkworth, Hannah M. Bailey, Margot E. Quinlan and Don van Meyel

Dear Don,

I am happy to tell you that your manuscript has been accepted for publication in Development, pending our standard publication integrity checks.

Reviewer 1

Advance summary and potential significance to field

Through elegant time-lapse imaging and molecular genetic analyses, this study demonstrates that Spir promotes actin polymerization at nascent branch points, leading to new branch formation, and that this pathway cooperates with Rab11-mediated secretory trafficking to establish the highly-branched arbor structure of Type IV da neurons. Overall, the manuscript presents an important advance linking Spir's cytoskeletal function to dendrite branch initiation and growth in vivo.

Comments for the author

The authors present new experimental data to address my major concern, and have thoroughly addressed other comments. The revised manuscript is improved. I congratulate the authors on this interesting, and elegantly executed study.

Reviewer 2

Advance summary and potential significance to field

The authors have adequately addressed my previous concerns and have significantly improved their work showing the role of the actin nucleator spire in dendrite branch initiation. In this form I believe the work provides interesting insight into part of the cytoskeletal machinery required dendrite morphogenesis.

Comments for the author

no further suggestions, but just a comment that in some analyses the variance of dendrite parameters seems quite high, which is unusual as c4da neuron dendrites are quite stereotyped and typically display fairly low variance if the animals are reared and staged under optimal conditions. The authors might want to review their staging and imaging protocol for future work to improve conditions (e.g. control staging timing, e.g. 3-4h max, and larval density in the food to optimize growth conditions, plus imaging at the same stage, e.g. at 96h±4h)